# Gaps in Implementing Bidirectional Screening for Tuberculosis and Diabetes Mellitus in Myanmar: An Operational Research Study

**DOI:** 10.3390/tropicalmed5010019

**Published:** 2020-02-01

**Authors:** Tun Kyaw Soe, Kyaw Thu Soe, Srinath Satyanarayana, Saw Saw, Cho Cho San, Si Thu Aung

**Affiliations:** 1National Tuberculosis Programme, Ministry of Health and Sports, Nay Pyi Taw 15012, Myanmar; drchochosanmph@gmail.com (C.C.S.); dr.sta.ntp@gmail.com (S.T.A.); 2Department of Medical Research (Pyin Oo Lwin Branch), Ministry of Health and Sports, Pyin Oo Lwin 05081, Myanmar; kyawthusoe.dmr@gmail.com; 3Center for Operational Research, The Union, 57006 Paris, France; ssrinath@theunion.org; 4Department of Medical Research (Head Quarter), Ministry of Health and Sports, Yangon 11191, Myanmar; sawsawsu@gmail.com

**Keywords:** tuberculosis, diabetes, random blood sugar, implementation, SORT IT, Myanmar, operational research

## Abstract

In Myanmar from July 2018, as a pilot project in 32 townships, all tuberculosis (TB) patients aged ≥40 years were eligible for diabetes mellitus (DM) screening by random blood sugar (RBS) and all DM patients attending hospitals were eligible for TB screening. We assessed the bidirectional screening coverage of target groups through a cross sectional study involving secondary analysis of routine program data. From January to March 2019, of the 5202 TB patients enrolled, 48% were aged ≥40 years. Of those aged ≥40 years, 159 (6%) were known to have DM, and the remaining 2343 with unknown DM status were eligible for DM screening. Of these, 1280 (55%) were screened and 139 (11%) had high RBS values (≥200 mg/dL, as defined by the national program). There was no information on whether patients with high RBS values were linked to DM care. Of the total 8198 DM patients attending hospitals, 302 (3.7%) patients were tested for sputum smear and 147 (1.7%) were diagnosed with TB. In conclusion, only half of the eligible TB patients were screened for DM and the yield of TB cases among screened DM patients was high. There is an urgent need for improving and scaling up bidirectional screening in the country.

## 1. Introduction

Globally, tuberculosis (TB) is one of the top 10 causes of deaths and still remains a global public health challenge, mainly affecting people living in low and middle income countries. In 2017, it was estimated that 10 million people (range: 9.0–11.1 million) developed TB disease and 1.3 million people died due to TB [1].

Diabetes mellitus (DM) is a chronic metabolic disease characterized by high blood glucose levels and is an important risk factor for several cardiovascular diseases. According to the International Diabetes Federation (IDF), 425 million adults globally were living with DM in 2017 and this will rise to 629 million by 2045. About 79% of adults with DM were living in low and middle income countries [2]. The World Health Organization (WHO) projects that DM will be the 7th leading cause of death in 2030 [3].

People with DM have immune system dysfunctions that increases their risk of TB infection, development of TB disease, and adverse TB treatment outcomes when compared to people without DM [4,5]. DM is therefore adversely affecting the global TB control/elimination efforts, especially in countries in Asia and Africa which have the highest burden of both diseases [5,6].

In order to guide the countries affected by the dual burden of TB and DM, the World Health Organization (WHO) and the International Union Against TB and Lung Disease (The Union), developed a framework for TB–DM collaborative activities in 2011 [7]. Under this framework, one of the key recommended activities is to establish mechanisms for collaboration between the public health programs that are providing TB and DM diagnostic and treatment services and conduct bidirectional screening (i.e., routine screening of all TB patients for DM and all DM patients for TB) and link patients with coexisting diseases to appropriate care. A few countries in Asia and Africa have formulated policies and are implementing bidirectional screening [8,9].

Myanmar (a country in Southeast Asia) is one of the 30 high TB burden countries in the world [1]. The prevalence of DM in the country is showing an increasing trend. According to the nationwide STEPS survey in 2014, which was based on the WHO Step-wise approach to surveillance of non-communicable disease methodology, the prevalence of DM among adults (aged 25–65 years) was 10.5% [10]. A hospital-based study conducted in Myanmar reported that the proportion of TB among DM patients was 8% [11].

TB control is a top priority in Myanmar’s National Health Plan (2016–2020). The National TB Programme (NTP) has developed a National Strategic Plan for Tuberculosis (2016–2020) in line with the WHO’s “The END TB Strategy” [12]. As per this plan, NTP in collaboration with the National Noncommunicable Disease (NCD) Control Programme has formulated guidelines for bidirectional screening of TB and DM patients and their linkage to care [13]. In July 2018, a pilot project was initiated in 30 townships and two tertiary hospitals with DM clinics across the country to assess the feasibility of implementing the guidelines prior to national scale-up. In this operational research study, we assessed extent of coverage of the target group with bidirectional screening six months after the commencement of the provision of screening in these pilot townships. We envisaged that the findings from this research will inform the NTP about any gaps in the implementation of the bidirectional screening guidelines and provide information on who is screened and not screened. This would providing valuable evidence for improving the guidelines and addressing any patient level challenges in expanding the bidirectional screening component to the rest of the country. 

Our study had two main objectives. In the 30 townships and two tertiary hospitals implementing the guidelines on bi-directional screening: (1) to determine the number and proportion of eligible TB patients screened for DM and assess the differences in the demographic and clinical characteristics of eligible TB patients screened for and not screened for DM; and (2) to determine the number of DM patients that were investigated for TB and diagnosed with TB. 

## 2. Materials and Methods

### 2.1. Study Design

This was a cross sectional descriptive study involving secondary analysis of routine program data.

### 2.2. Setting 

#### 2.2.1. NTP

The NTP under the Department of Public Health, Ministry of Health, and Sports is functioning with 17 regional and state TB centers with 404 TB teams at district and township levels. TB control activities are being implemented at township levels by the township medical officer, through integration with primary health care. NTP covered all 330 townships with Direct Observed Treatment Strategy (DOTS) in November 2003 and covered the five new townships established in the Nay Pyi Taw Union Territory in 2011.

People with TB symptoms attending the public health facilities are identified and diagnosed using a diagnostic algorithm that consists of a combination of diagnostic tools (sputum smear microscopy, chest radiography, and Xpert MTB/Rif assay test [14]. All diagnosed TB patients as per their disease status are categorized and treated with standard regimens of first line or second line anti-TB drugs. The disease classification, treatment regimens, recording, and reporting of treatment outcomes is as per the international guidelines recommended by the WHO [14].

#### 2.2.2. NCD Control Programme

The Myanmar’s NCD Control Programme (initiated in early 2015) consists of medical services for DM, hypertension, heart disease, stroke, chronic lung disease, and cancers. The programme has developed guidelines for screening, diagnosis, and management of patients with these diseases. The NCD programme services are implemented through public health facilities. DM management services are provided through the NCD clinics that are established at the township and rural health centers. At three tertiary health facilities, there are exclusive DM clinics. 

#### 2.2.3. TB/DM Collaborative Activity

Myanmar’s NTP and NCD Control Programme developed guidelines for TB–DM collaborative activities in 2017. The overall objectives of the guidelines are to (a) establish collaborations between TB and NCD control programmes at the national, state, district, and township levels, and (b) screen TB patients for DM and DM patients for TB and link patients with coexisting diseases to appropriate care. The implementation of these guidelines was initiated as a pilot project in 30 townships and two tertiary hospitals in July 2018. The 30 townships were selected purposively by NTP to include at least one township in each state/region in the country.

#### 2.2.4. Screening Process

All TB patients aged 40 years and above are asked about preexisting DM by the township TB coordinator at the time of enrolment for TB treatment. If they do not have a history of DM, the TB coordinator or laboratory technician screens them for DM by measuring their random capillary blood sugar (RBS) levels using a glucometer. If the RBS is more than 200 mg/dL then they are evaluated for the presence of DM symptoms (polyuria, polydipsia, polyphagia, and unexplained weight loss). If they are positive for any of the DM symptoms, they are referred to NCD/DM clinics for diagnosis and management by using a referral form. If the DM symptoms are not present, the patients are subjected to another RBS or fasting blood glucose (FBS) test within a week. If the RBS is >200 mg/dL or if the FBS is >126 mg/dL, then the patient is referred to the DM/NCD clinics for further management. Those diagnosed with DM are managed as per the NCD programme guidelines [15]. 

For detecting TB among DM patients, all DM patients attending outpatient department (OPDs) of hospitals or NCD/DM clinics are screened for TB symptoms (low grade fever, cough > 2 weeks, weight loss, night sweats, chest pain, enlarged neck gland). If the patients have at least one of these TB symptoms, then they are considered as presumptive TB and referred to a TB clinic. DM patients with presumptive TB are investigated using sputum smear microscopy for Acid Fast Baccili (AFB) and chest radiography at the TB clinic. They are also referred to an Xpert MTB/Rif assay facility for a drug susceptibility test if their sputum smear test is positive or if they are clinically diagnosed as having TB. DM patients diagnosed with TB are treated with standard anti-TB treatment as per the NTP’s TB treatment guidelines [13].

#### 2.2.5. Recording and Reporting

The screening of TB patients for DM is recorded in the township TB register, on a treatment card, and in the township laboratory registers by the NTP team. The screening of DM patients for TB is recorded in the OPDs registers at general hospitals or DM patient registers at DM clinics. Referral/feedback forms are used to cross refer patients and obtain the evaluation status of the patients. These referral/feedback forms are maintained by the staff of the TB and NCD Control Programme. On a quarterly basis, the staff of the NTP and NCD programme at the township level compile and report data on the number of patients who underwent bidirectional screening and the results of this screening to the regional centers. At the regional centers, the data from various townships are compiled and the compiled reports are sent to the NTP central unit located at Nay Pyi Taw.

### 2.3. Study Patient Population and Study Period

For objective (1), we included all TB patients enrolled for TB treatment in 30 townships in Myanmar from January to March 2019. For objective (2), we included all DM patients attending general OPD or DM clinics who were screened for TB at the 30 pilot townships and two tertiary hospitals with DM clinics in Myanmar for the same period. All persons included in the study were all residents of these townships. 

### 2.4. Data Variables, Sources of Data, and Data Collection

For objective 1, we collected individual patient data of TB patients, which included the name of states/regions, the name of the townships, the patients TB register number, the patient’s age, sex, and type of TB, the site of TB, sputum smear and GeneXpert result at the time of diagnosis, type of regimen, HIV, ART, CPT, and DM status, whether they were RBS tested or not, and their RBS result. These data were abstracted from township TB registers. For objective 2, we collected the aggregate number of total DM patients attending OPDs, DM patients who were tested for TB, and patients diagnosed as having TB from quarterly township TB–DM reports. Photo copies of township TB registers and reports were obtained from respective township and hospitals to facilitate the data collection process.

### 2.5. Data Entry and Analysis

Data from the hard copies were entered into a structured proforma created in EpiData Entry (Version 3.1). We analyzed the data using EpiData analysis (Version 2.2.2.183, EpiData Association, Odense, Denmark) and Stata (version 15.1, Stata Corp, College Station, TX, USA).

For objective 1, we have provided a summary of the number of TB patients registered for treatment in the 30 townships, the number (and proportion) of TB patients that were eligible for DM screening, the number (and proportion) who underwent DM screening (RBS test), and the number (and proportion) who screened positive for RBS (RBS > 200 mg/dL). 

We have also summarized the demographic and clinical variables of the DM screening eligible TB patients by using numbers (and proportion). We disaggregated the DM screening eligible population into those who underwent screening (i.e., had an RBS test) and those who did not undergo screening (i.e., those without an RBS test). We assessed the independent association between the demographic and clinical characteristics with screening/not screening using binomial log models after adjusting for clustering at the township/regional level and have presented the associations using risk ratios and adjusted risk ratios. 

For objective 2, we provided a summary table containing aggregate numbers of DM patients who attended the OPD clinics/DM clinics at 30 townships and two tertiary hospitals, the number and proportion who underwent TB tests, and the number and proportion who diagnosed with TB. 

### 2.6. Ethics Approvals

Permission for the study was granted by the Institutional Review Board of Department of Medical Research (Ethics/DMR/2019/090) and the Ethics Advisory Group of The Union, Paris, France (08/19).

## 3. Results

### 3.1. DM Screening Among TB Patients

Data were available from 30 pilot townships. In these townships, 5202 TB patients were enrolled for TB treatment from January to March 2019. Of those enrolled, 2502 (48%) were aged ≥40 years. Of those aged ≥40 years, 159 (6%) were already known to have DM. The remaining 2343 TB patients with unknown DM status were eligible for DM screening. Of these, 1280 (55%) were screened for DM by RBS and 139 (11%) had RBS values of ≥200 mg/dL (Figure 1). The proportion screened across the 30 townships ranged from 0% to 100%. There was no information in the recording and reporting formats on whether all TB patients with high RBS values had symptoms of DM, what proportion underwent subsequent testing fasting blood sugar ((FBS) and/or glycosylated hemoglobin) and were diagnosed with DM and linked to DM care. 

The demographic and clinical characteristic of 2343 TB patients eligible for DM screening are described in Table 1. Their mean (SD) age was 56.5 (11.3) years, 1569 (67%) were male, 2015 (86%) were new TB patients, 2030 (87%) had pulmonary TB and 167 (7%) were HIV positive. The association between these patients’ demographic and clinical characteristics with nonscreening for DM by RBS testing is given in Table 2. Patients who were positive for rifampicin resistance were less likely to be screened for DM. Apart from this, no other characteristic was associated with DM screening.

### 3.2. TB Screening Among DM Patients

In the 30 townships and two tertiary hospitals with DM clinics, 8198 DM patients were reported to have attended the DM clinics of which 2686 (33%) were male. There was no information on how many of these DM patients were systematically screened for TB symptoms. However, there was information on the number who underwent sputum smear examination, the number diagnosed with TB, and the number diagnosed with bacteriologically confirmed TB disaggregated by gender. This information is given in Table 3. Of the total DM patients, 302 (3.7%) patients were tested for sputum smear (6.2% among males and 2.4% among females). Overall, 147 (1.7%) were diagnosed with TB (3.1% among males and 1.1% among females) with 89 (1.1%) patients being bacteriologically confirmed TB (2% among males and 1.1% among females). All diagnosed TB patients were linked to TB care.

## 4. Discussion

This is the first study in Myanmar to assess the implementation status of bidirectional screening for TB and DM. The study has three major findings. First, nearly half of the TB patients were eligible for DM screening. Second, of those eligible, only half were screened for DM by RBS and of those screened ~10% had high blood glucose values. Third, about 4% of the DM patients had undergone sputum smear examination and 2% had TB.

The major strength of the study was that it was done using data collected under routine programmatic conditions from more than 90% of the pilot sites and therefore it reflects ground level reality. 

The major limitations of the study are that we used data recorded in programmatic records and the results may be incorrect if there are any deficiencies or errors in recording and reporting. The NTP has strong supervision and monitoring systems with frequent checks at all levels, and therefore we feel that the gaps noted in this study are likely to be due to deficiencies in implementation rather than due to errors in recording and reporting. We also conducted the study in the first few months of the roll out of bidirectional screening and therefore the study findings reflect gaps in early implementation of the bidirectional screening. We feel that our study provides baseline information against which future comparisons may be made. The data analysis is restricted to only those variables that are routinely recorded in program records. There can be several unmeasured patient and health system level factors (e.g., socioeconomic status of the patients or availability of glucometers) responsible for the observed gaps. Since these were unmeasured in our study, we are unable to account for these factors in our analysis or data interpretation. Finally, due to resource constraints, we did not include a qualitative study component (by interviewing health care providers or patients) to explain the study results. Therefore, the exact reasons for the gaps in implementation are unknown.

Despite these limitations, the study findings have the following implications on policy and practice.

First, nearly half of the TB patients enrolled in the pilot townships were eligible for DM screening. This is low when compared to other neighboring countries such as India and China which use a different eligibility criterion (e.g., age > 30 years) [16,17]. Depending on the age distribution of the TB patients and prevalence of DM in various age groups and the availability of resources, Myanmar can consider either lowering or increasing the age-related eligibility criteria for DM screening among TB patients. Our study finding provides background information for undertaking such studies or decisions in the future. 

Second, of those TB patients who were eligible, only half (55%) were screened for DM. This is lower than studies from other settings such as India and China where more than 90% of the eligible patients had undergone the screening [18,19]. In our study, we found huge variations in the proportion screened in the 30 townships (0–100%). This indicates an urgent need for assessing the reasons for non-screening and suboptimal screening in certain townships. Other than this, our study findings show that a lower proportion of TB patients with rifampicin resistance had undergone the DM screening. The exact reasons why patients with rifampicin resistance were not screened is unknown. Non-screening of DM in TB patients with rifampicin resistance is a missed opportunity for providing optimal care to this subgroup of patients. 

Third, about 11% of the TB patients screened had RBS values >200 mg/dL warranting further evaluation for DM diagnosis. Unfortunately, there was no information on whether these patients had symptoms of DM and/or whether these patients reached DM/NCD clinics, whether they underwent further DM diagnostic evaluation, or what their DM status was. This was due to deficiencies in recording and reporting systems which did not outline mechanisms to capture this component of care linkage in the TB–DM bidirectional screening. In order to strengthen linkages to DM diagnosis and treatment in TB patients, the recording and reporting system needs to be revised or updated to capture the information in this care cascade. 

Fourth, in order to identify DM among TB patients, Myanmar is using an RBS cutoff value of >200 mg/dL with DM symptoms. This cutoff value is highly specific but relatively less sensitive for identifying persons with DM [20]. This raises concerns about missing DM among TB patients under routine program conditions. The current international guidelines by The Union recommends that TB patients with RBS of >110 mg/dL undergo FBS or glycosylated hemoglobin testing for confirmation of DM diagnosis [21]. Myanmar may review these new guidelines and consider adopting them. 

Lastly, nearly 4% of DM patients had undergone sputum smear examination and 1.7% were diagnosed with TB. The yield (1.7%) of TB among DM patients is high when compared to what has been reported in other countries [18]. However, we were unable to determine if there were any systematic differences in the DM patients selected for screening for TB and those not screened, and thus whether the relatively high yield of patients positive for TB reflects the risk of the general DM population or of a particular at-risk subgroup. 

In conclusion, data from the TB–DM bidirectional screening pilot project in Myanmar shows that there are several gaps in screening and linkage to care. These gaps in screening varied across townships. The deficiencies identified can be addressed by improving the recording and reporting system and by conducting qualitative research to identify context specific reasons and interventions to rectify the identified issues.

## Figures and Tables

**Figure 1 tropicalmed-05-00019-f001:**
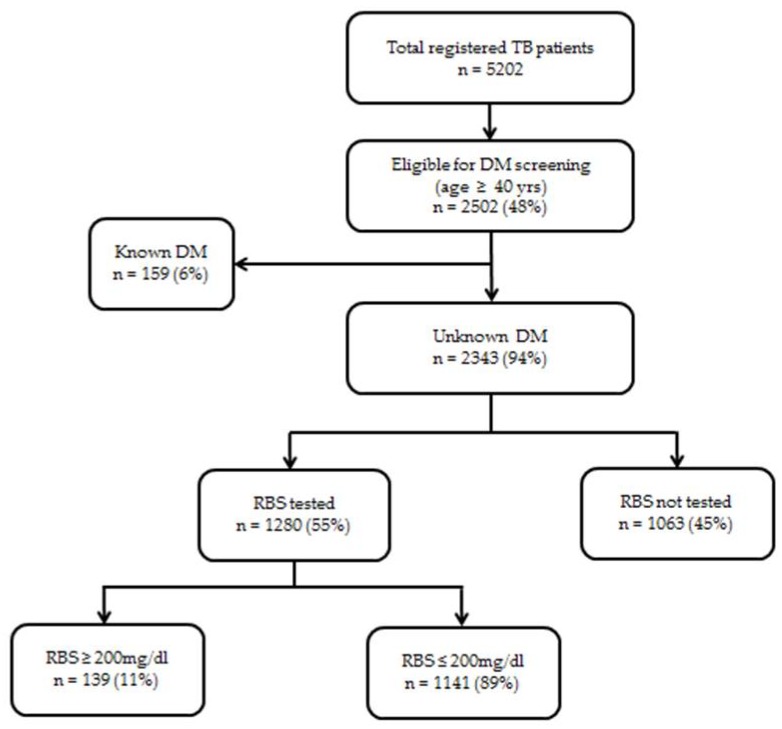
Flow of diabetes mellitus screening among registered tuberculosis patients aged ≥40 years in 30 pilot townships, Myanmar, from January to March, 2019. TB = tuberculosis; DM = diabetes mellitus, RBS = random blood sugar.

**Table 1 tropicalmed-05-00019-t001:** Demographic and clinical characteristics of all tuberculosis patients eligible for DM screening in 30 pilot townships, Myanmar, from January to March 2019.

Characteristics	Number Eligible	
	*n*	(%)
**Total**	2343	100.0
**Age**		
40–49	755	32.2
50–59	676	28.9
60–69	565	24.1
≥70	347	14.8
**Gender**		
Male	1569	67.0
Female	774	33.0
**Type of TB Patients**		
New	2015	86.0
Previously treated	283	12.1
Unknown/not recorded	45	1.9
**Site of TB**		
Pulmonary TB	2030	86.6
Extrapulmonary TB	162	6.9
Not recorded	151	6.4
**Sputum Result at Diagnosis**		
Negative	1228	52.4
Positive	965	41.2
Not recorded	150	6.4
**GeneXpert Result at Diagnosis**		
MTB not detected	686	29.3
MTB detected Rifampicin Sensitive	1121	47.8
MTB detected Rifampicin Resistant	16	0.7
Not recorded	520	22.2
**Treatment Regimen**		
Initial regimen	2020	86.2
Retreatment regimen	171	7.3
Second Line anti TB drug	8	0.3
Not recorded	144	6.1
**HIV, CPT, ART**		
HIV negative	2026	86.5
HIV positive and on CPT and ART	126	5.4
HIV positive and not on either CPT and/or ART	41	1.7
HIV unknown/not recorded	150	6.4

DM = diabetes mellitus, TB = tuberculosis, MTB = mycobacterium tuberculosis, AFB = acid fast bacilli, HIV = human immunodeficiency virus, ART = anti-retro viral treatment, CPT = cotrimoxazole prophylaxis therapy.

**Table 2 tropicalmed-05-00019-t002:** Factors associated with not being screened for diabetes mellitus among eligible tuberculosis patients with unknown DM in 30 pilot townships, Myanmar, from January to March 2019.

Characteristics	Number Eligible	Not Screened for DM		Relative Risk	Adjusted Relative Risk	*p*-Value
		*n*	%	(95% CI)	(95% CI)	
**Total**	2343	1063	45.4			
**Age**						
40–49	755	347	46.0	Reference	Reference	
50–59	676	303	44.8	0.97 (0.82–1.14)	0.97 (0.84–1.12)	0.744
60–69	565	257	45.5	0.98 (0.81–1.20)	0.96 (0.82–1.13)	0.665
≥70	347	156	45.0	0.97 (0.80–1.18)	0.95 (0.80–1.13)	0.614
**Gender**						
Male	1569	693	44.2	Reference	Reference	
Female	774	370	47.8	1.08 (0.98–1.18)	1.05 (0.96–1.15)	0.219
**Type of TB Patients**						
New	2015	923	45.8	Reference	Reference	
Previously treated	283	116	41.0	0.89 (0.65–1.21)	0.85 (0.62–1.16)	0.325
Unknown/not recorded	45	24	53.3	1.16 (0.78–1.73)	1.15 (0.80–1.66)	0.432
**Site of TB**						
Pulmonary TB	2030	951	46.8	Reference	Reference	
Extrapulmonary TB	162	70	43.2	0.92 (0.67–1.26)	0.89 (0.66–1.19)	0.458
Not recorded	151	42	27.8	0.59 (0.22–1.59)	0.49 (0.13–1.78)	0.281
**Sputum Result at Diagnosis**						
Negative	1228	569	46.3	Reference	Reference	
Positive	965	421	43.6	0.94 (0.83–1.05)	0.90 (0.74–1.08)	0.267
Not recorded	150	73	48.7	1.05 (0.50–2.18)	1.21(0.72–2.03)	0.464
**Xpert Result at Diagnosis**						
MTB not detected	686	327	47.7	Reference	Reference	
MTB detected Rifampicin Sensitive	1121	499	44.5	0.93 (0.81–1.07)	0.97 (0.86–1.10)	0.722
MTB detected Rifampicin Resistant	16	12	75.0	1.57 (0.92–2.68)	1.62 (1.07– 2.46)	0.022
Not recorded	520	225	43.3	0.90 (0.53–1.53)	0.86 (0.52–1.41)	0.558
**HIV, CPT, ART**						
HIV negative	2026	902	44.5	Reference	Reference	
HIV positive and on CPT and ART	126	55	43.7	0.98 (0.79–1.20)	0.92 (0.75–1.130)	0.448
HIV positive and not on either CPT and/or ART	41	24	58.5	1.31 (0.87–1.98)	1.27 (0.88–1.83)	0.2
HIV unknown/not recorded	150	82	54.7	1.22 (0.48–3.08)	1.45 (0.80–2.62)	0.213

DM = diabetes mellitus, TB = tuberculosis, MTB = mycobacterium tuberculosis, AFB = acid fast bacilli, HIV = human immunodeficiency virus, ART = anti-retro viral treatment, CPT = cotrimoxazole prophylaxis therapy.

**Table 3 tropicalmed-05-00019-t003:** Tuberculosis screening of diabetes mellitus patients in 30 pilot townships and two tertiary hospitals with DM clinics, Myanmar, from January to March 2019.

Variable	Male	Female	Total
	*n*	*n*	*n*
Total DM patients	2686	5512	8198
DM patients who were tested by sputum smear	167 (6.2%)	135 (2.4%)	302 (3.7%)
DM patients who were diagnosed with TB	84 (3.1%)	63 (1.1%)	147 (1.7%)
DM patients with bacteriologically confirmed TB	55 (2.0%)	34 (0.6%)	89 (1.1%)

DM = diabetes mellitus, TB = tuberculosis.

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
