# Peer review of "Gaps in Implementing Bidirectional Screening for Tuberculosis and Diabetes Mellitus in Myanmar: An Operational Research Study"

_tropicalmed, 2020, doi:10.3390/tropicalmed5010019_

Round 1
Reviewer 1 Report
The authors report the results of a cross sectional study aimed at assessing the implementation status of the guidelines on bi-directional screening (screening of all tuberculosis -TB, patients for diabetes mellitus –DM, and all DM patients for TB) in Myanmar.
The screening is included in the TB-DM collaborative activities developed by the Myanmar’s National TB Program (NTP) and Non-communicable Diseases (NCD) control programme.
The study was carried out in 30 pilot townships (DM screening among TB patients and TB screening among DM patients) and two tertiary hospitals with DM clinics (only TB screening among DM patients).
Although the study presents a number of limitations, well addressed in the Discussion, the obtained results clearly evidence some gaps in the implementation of the bi-directional screening.
Such research studies, conducted in the early phase of implementation of the screening are very important to provide information for undertaking further studies or decisions in the future.
Very minor changes:
Line 59: Non-communicable diseases should be add before NCDs
Line 242: TB screening should be DM screening
Line 263 the strengthen should be to strengthen
Author Response
Dear Reviewer
Please see the attachment.
with regards
Tun Kyaw Soe

Reviewer 2 Report
The paper from Soe et al examines the gaps in implementing bidirectional screening for TB and DM in Myanmar. The study is an observational and descriptive one, hence the lower originality; though the importance is high. the paper is well-written and presented. This reviewer has only a few minor comments, as follows:
1. The authors could include the short period of analysis (3 months Jan to March, 2019) as a limitation of the study, i.e. it provides a brief snapshot of the current position.
2. Ethics approval - the authors should clarify. Permission was 'sought', but has it actually been provided? If yes, then change this to say that permission was provided by the IRB.
3. In Line 171, what is the population of these townships, to get an idea of the TB rates?
4. Line 184 (and discussion): can the authors speculate on why patients with rifampicin resistance were less likely to be screened for DM?
Author Response
Dear Reviewer Please see the attachment. With regards Tun Kyaw Soe

Reviewer 3 Report
This a preliminary study of a pilot program in its early stages. Although the title suggests that this is an ‘operational research’ study, and the text refers to ‘assess the implementation status’, the study is actually an assessment of the extent of coverage of the target group with screening. As the authors note, further qualitative research is needed to understand the reasons for the low coverage.
This is also complicated by lack of clarity on the extent of implementation. The paper refers to the pilot process commencing in July 2018, and that the assessment was conducted in January-March 2019, but it is not clear whether the commencement refers to operational commencement (ie following training, provision of resources etc), or commencement of the introduction of the screening.
There is no description of how the townships were selected which means it is not possible to know how representative this might be of townships in Myanmar; or what population was covered, given the very significant differences between different regions in Myanmar, not just in population characteristics, but also in social, economic and security status.
Perhaps this is more appropriate as a short report rather than a full article; given the lack of significant differences between screened and non screened TB patients, the full data in table 2 is probably not required.
Abstract
Page 1 line 16 ‘Assess the implementation status’ – this statement is not very clear or easy to understand. It would appear that an assessment of the proportion eligible for screening who were reported as receiving screening was done. This is more an assessment of the coverage of the screening rather than implementation.
Line 24 the high yield was only among those DM patients screened – not among all DM patients
Introduction
Page 2 line 64 – ‘assess the implementation status’ – as noted above is not clear. Suggest provide a more specific term eg assess the extent of screening coverage of the eligible population
It would be useful to indicate the date the assessment was undertaken in relation to the commencement of the screening program. This is noted as January to March 2019 in line 172 ie about 6 months after the commencement of the program in July 2018. It would be useful to provide some description of the extent to which training and resources had been completed by the time of the assessment. Clarify whether the commencement of the pilot meant the commencement of training and resource provision; or was this following provision of training and resources ? If the July date was the commencement of training etc, then the assessment after 6 months may be rather early.
Page 2 line 70 – the rationale for the comparison between screened and non screened eligible TB patients is not clear. Was there some hypothesis that there might be differences ? Clarify why this comparison was undertaken. Eg was there a possibility that patients thought to be at higher risk of DM would be more likely to be screened ?
Materials and Methods
The study population is not clearly defined, and the relationship between the townships and the referral hospitals is not clear. Are the study population members resident of the selected towns ? or are they just those who sought treatment from the selected townships ?
There is further information on page 4 lines 135 – 139 but the relationship between the 30 townships and the referral hospital remains unclear. For example it may be that the referral hospital patients come from a different population (eg urban, wealthier) than the township patients. Facilities at the referral hospitals may also be better than at townships and may result in differing rates of screening. Justification is needed for combining these two populations.
More information is needed on how the 30 townships were selected, what is the distribution of townships across Myanmar, and how representative this sample might be of the situation nationally.
Results
Discussion
Page 8 line 228 – is should be are
Line 229 reflects on the points above on the timing of the assessment in relation to the process of piloting. See comments above.
Page 8 line 237 The lack of measurement of the process of implementation is a major weakness in the study. This raises the question as to whether this can be called ‘implementation research’ when it does not address the process of implementation.
Page 8 line 242 - ? should be diabetes screening ?
Page 9 line 252 – This is an important finding and should be reported with the data in the results section. It is also relevant to the selection of townships – which, as noted above, is not described.
Page 9 line 273 – potential that those at greater risk / suspected TB screened resulted in higher TB positive among screened DM patients
Conclusion
Given the importance of this early assessment for the program, more significant findings could be included in the conclusion eg the extent of variation in proportion screened among different townships; and the very low apparent rate of TB screening of DM patients. This suggests that more than improving the recording and reporting system is required.
Author Response
Dear Reviewer
Please see the attachment.
With Regards
Tun Kyaw Soe

Round 2
Reviewer 3 Report
I have added my comments / respnoses to the author's in the cover letter below in bold:
I have added my comments / responses in bold to the letter from the authors below: Where the authors have adequately responded to the orginal comment I have noted 'addressed'.
Re: Gaps in implementing bidirectional screening for Tuberculosis and Diabetes Mellitus in Myanmar: An operational research study [Manuscript ID: tropicalmed-619838]
Dear Editor,
We thank you and the reviewers for your comments and suggestions to improve the content and clarity of the manuscript. We have made several revisions in the manuscript as per these suggestions. We also provide a point by point response to each of the reviewers’ in this rebuttal letter. We have highlighted our responses/changes in yellow color the rebuttal letter and in the revised manuscript. We hope that you will find them satisfactory. If there are any further suggestions to improve the content or clarity of the manuscript, please feel free to write to us. We will be happy to incorporate them into our manuscript.
Tun Kyaw Soe
(On Behalf of all the co-authors)
Reviewer 3
This a preliminary study of a pilot program in its early stages. Although the title suggests that this is an ‘operational research’ study, and the text refers to ‘assess the implementation status’, the study is actually an assessment of the extent of coverage of the target group with screening. As the authors note, further qualitative research is needed to understand the reasons for the low coverage.
Authors’ response: Many thanks for your comment. We agree that the phrase “assess the implementation status” is a non-specific term and is ambiguous. We had cleared this ambiguity by providing specific objectives in the last paragraph of introduction in the main manuscript. We have now used the term ‘coverage’ in the abstract and in the introduction. We hope this addresses the concern of the reviewer.
Reviewer: addressed
This is also complicated by lack of clarity on the extent of implementation. The paper refers to the pilot process commencing in July 2018, and that the assessment was conducted in January-March 2019, but it is not clear whether the commencement refers to operational commencement (ie following training, provision of resources etc), or commencement of the introduction of the screening.
Authors’ response: We have now clarified in the revised manuscript that the patient screening process was initiated from July 2018 onwards. This happened after conducting essential trainings and providing necessary resources (line 62-63).
Reviewer: This could be made clearer by shifting the phrase ‘after conducting essential trainings and providing necessary resources’ from line 61 ff to refer to the timing of the study, rather than the timing of the project project – to follow line 65 as ‘we assessed the extent of coverage of the target groups with bidirectional screening 6 months after the commencement of the provision of screening in the pilot project sites’.
There is no description of how the townships were selected which means it is not possible to know how representative this might be of townships in Myanmar; or what population was covered, given the very significant differences between different regions in Myanmar, not just in population characteristics, but also in social, economic and security status.
Authors’ response: Thank you for pointing this out. We have clarified in the manuscript, that the “The 30 townships were selected purposively by NTP to include at least one township in each State/Region in the country” (line 107-108)
Reviewer: addressed
Perhaps this is more appropriate as a short report rather than a full article; given the lack of significant differences between screened and non-screened TB patients, the full data in table 2 is probably not required.
Authors’ response: We slightly disagree with the reviewer here. We feel that we have to present the study results as per the protocol and not based on whether the study results were significant or not and in an operational /programme setting, non-significant results are as important as significant results. We wish to retain table 2 and present this as a full research article by presenting the results as per the protocol that was approved by the ethics committee. We feel that it is inappropriate to condense the study into a short report and present only the significant findings. We hope that the reviewer agrees to our point of view.
Reviewer: This is perhaps a question more relevant to the editors as to whether the journal wishes to publish the full report. I agree that it is important to report studies where non-significant differences are found and with the changes following review, the paper is clearer on its purpose.
Abstract
Page 1 line 16 ‘Assess the implementation status’ – this statement is not very clear or easy to understand. It would appear that an assessment of the proportion eligible for screening who were reported as receiving screening was done. This is more an assessment of the coverage of the screening rather than implementation.
Authors’ Response: Thank you very much. We have now used the phrase “bidirectional screening coverage of target groups” (Line 15-16).
Reviewer: addressed
Line 24 the high yield was only among those DM patients screened – not among all DM patients
Authors’ Response: Thank you. We have revised the sentence. (Line 24)
Reviewer: addressed
Introduction
Page 2 line 64 – ‘assess the implementation status’ – as noted above is not clear. Suggest provide a more specific term eg assess the extent of screening coverage of the eligible population.
Authors’ response: Many thanks for your comment. We have now revised the text to “we assessed extent of coverage of the target group with bidirectional screening in these pilot townships”. (Line 64-65)
It would be useful to indicate the date the assessment was undertaken in relation to the commencement of the screening program. This is noted as January to March 2019 in line 172 ie about 6 months after the commencement of the program in July 2018. It would be useful to provide some description of the extent to which training and resources had been completed by the time of the assessment. Clarify whether the commencement of the pilot meant the commencement of training and resource provision; or was this following provision of training and resources? If the July date was the commencement of training etc, then the assessment after 6 months may be rather early.
Response: Thank you very much. We have now clarified that patient screening was initiated in these townships from July onwards and the trainings and resource provision preceded this date and continued throughout (Line 62-63).
Reviewer: see comments and suggested changes above
Page 2 line 70 – the rationale for the comparison between screened and non-screened eligible TB patients is not clear. Was there some hypothesis that there might be differences? Clarify why this comparison was undertaken. Eg was there a possibility that patients thought to be at higher risk of DM would be more likely to be screened?
Response: Thanks for raising this issue. The reviewer is right, and we hypothesized that patients with certain characteristics may be left out from the screening process and we, therefore compared the characteristics of those screened and not screened. We have clarified this aspect in the introduction (Lines 66-70)
Reviewer: addressed
Materials and Methods
The study population is not clearly defined, and the relationship between the townships and the referral hospitals is not clear. Are the study population members resident of the selected towns ? or are they just those who sought treatment from the selected townships ?
Author Response: We have clarified that the study population included all registered TB patients in each pilot township. They were residents at each township. (Lines 142-143)
Reviewer: addressed.
There is further information on page 4 lines 135 – 139 but the relationship between the 30 townships and the referral hospital remains unclear. For example it may be that the referral hospital patients come from a different population (eg urban, wealthier) than the township patients. Facilities at the referral hospitals may also be better than at townships and may result in differing rates of screening. Justification is needed for combining these two populations. More information is needed on how the 30 townships were selected, what is the distribution of townships across Myanmar, and how representative this sample might be of the situation nationally.
Response: Thank you for highlighting this aspect. We have now clarified about the selection and distribution of the townships in our manuscript (Line 107-108). We wish to clarify that there was no relationship between the townships and the two referral hospitals. We pooled the data to protect the identity of the individual townships and referral hospitals. We had committed to the ethics committee that we will disseminate only aggregate pooled data and will not disaggregate the results township/hospital wise indicating good performers and bad performers. We agree that the characteristics of the patients in the township and the hospitals could different.
Reviewer: The additional sentence in line 107-108 has clarified the representativeness of the sample. I understand from the revisions that the study population for the TB screening were patients presenting at the 30 township clinics; while the study population for the diabetes screening combined those presenting at township clinics, and those presenting to the two referral hospitals. If this is not correct, please clarify in the text.
Discussion
Page 8 line 228 – is should be are
Authors’ response: Thank you very much. We have revised it. (line 231)
Reviewer: addressed
Line 229 reflects on the points above on the timing of the assessment in relation to the process of piloting. See comments above.
Authors’ response: Thank you. At the time of assessment, all pilot townships have received complete trainings.
Reviewer: addressed
Page 8 line 237 The lack of measurement of the process of implementation is a major weakness in the study. This raises the question as to whether this can be called ‘implementation research’ when it does not address the process of implementation.
Authors’ response: Agree. It is one of the limitations. We have therefore mentioned the need of qualitative research in conclusion to understand the challenges in the process of implementation. We feel that operational research or implementation research is a stepwise process. We can begin by examining any of the four aspects—inputs, processes, outputs, outcomes—in isolation or in combination and based on the results we can examine the other aspects.
Reviewer: addressed with use of the term operational research
Page 8 line 242 - ? should be diabetes screening ?
Authors’ response: Thanks. We have revised. (line 245)
Reviewer: addressed
Page 9 line 252 – This is an important finding and should be reported with the data in the results section. It is also relevant to the selection of townships – which, as noted above, is not described.
Authors’ response: We have mentioned this in the results section (line number 181-182). We have clarified about the selection of townships (Line 107-108)
Reviewer: addressed
Page 9 line 273 – potential that those at greater risk / suspected TB screened resulted in higher TB positive among screened DM patients
Authors’ response: Yes, it is possible that only those at greater risk were screened for TB and this resulted in higher TB positive. We have clarified (in line number 282-285) that since there was no documentation of whether everyone was systematically screened for TB we are unable to comment whether the screening process if functioning optimally or not.
Reviewer: I suggest rewording line 282 ff as below – the term ‘functioning optimally’ is very broad, while I think you can be more specific about your conclusion
Line 283 ‘However, we were unable to determine if there were any systematic differences in the DM patients selected for screening for TB and those not screened, and thus whether the relatively high yield of patient positive for TB reflects the risk of the general DM population, or of a particular at risk sub-group’.
Conclusion
Given the importance of this early assessment for the program, more significant findings could be included in the conclusion eg the extent of variation in proportion screened among different townships; and the very low apparent rate of TB screening of DM patients. This suggests that more than improving the recording and reporting system is required.
Authors’ response: Thank you for the suggestion. We have added a sentence that the gaps varied across townships. In addition to improving the recording and reporting system we have also highlighted the importance of qualitative research to understand and address the context specific challenges in implementation.
Reviewer additional comments
Line 143 – ‘all persons included in the study were residents of these townships’ (omit second all)
Line 208 I note that the title for Table 1 is confusing. It states ‘eligible TB patients… who screened for and not screened for DM’ – but the table does not include screening information. Revise title to ‘all TB patients eligible for DM screening’
Author Response
Dear Reviewer,
Please see the attachment.
with regards,
Tun Kyaw Soe
